# THE MAJORITY IS NOT ALWAYS RIGHT: RL TRAINING FOR SOLUTION AGGREGATION

## ABSTRACT

Scaling up test-time compute, by generating multiple independent solutions and selecting or aggregating among them, has become a central paradigm for improving large language models (LLMs) on challenging reasoning tasks. While most prior work relies on simple majority voting or reward model ranking to aggregate solutions, these approaches may only yield limited benefits. In this work, we propose to learn aggregation as an explicit reasoning skill: given a set of candidate solutions, we train an aggregator model to review, reconcile, and synthesize a final, correct answer using reinforcement learning from verifiable rewards. A key ingredient is careful balancing of easy and hard training examples, allowing the model to learn both to recover minority-but-correct answers as well as easy majority-correct answers. Empirically, we find our method, AGGLM, outperforms both strong rule-based and reward-model baselines, across multiple benchmarks. Furthermore, it generalizes effectively to solutions from differing models, including stronger ones than contained in the training data, all while requiring substantially fewer tokens than majority voting with larger numbers of solutions.

## 1 INTRODUCTION

Scaling up test-time compute, by producing longer intermediate thoughts and/or selection or aggregation over multiple generated samples, has become a predominant way to improve large language models (LLMs) on challenging reasoning tasks (Wei et al., 2022; Wang et al., 2023; Brown et al., 2024). These strategies trade additional inference-time computation for higher accuracy, and have yielded state-of-the-art results across mathematics, code generation, and scientific problem solving (Jaech et al., 2024; Guo et al., 2025).

Making use of multiple independent solutions has the advantage of parallel computation, but leaves open the central design choice of how to aggregate answers. Standard practice is to either apply majority voting over the sampled solutions, or weighted majority voting, optionally guided by reward models or verifiers to select among candidates (Wang et al., 2023; Brown et al., 2024; Wu et al., 2025). However, majority-based aggregation can overlook valuable minority solutions: correct answers are sometimes assigned low probability under the model due to modeling errors (Stahlberg & Byrne, 2019; Stahlberg et al., 2022). Hence, simply maximizing model score, or picking the most frequent output, will not yield the optimal result (Fu et al., 2025). Furthermore, these methods leave unexploited the partial correctness that can exist *within* otherwise incorrect thought traces, missing opportunities to combine correct steps from different candidates to produce a fully correct solution.

This paper proposes AGGLM, a method that instead *learns to use reasoning to combine thoughts* produced by multiple generations computed in parallel. AGGLM is simple: (i) sample multiple solutions from an LLM; (ii) pass these solutions back to an LLM with an aggregation instruction that asks it to synthesize a final answer by reconciling, correcting, and combining the intermediate reasoning steps. We then train a reasoning model to perform this aggregation using reinforcement learning from verifiable rewards (RLVR). This turns aggregation itself into a learned reasoning skill rather than a fixed heuristic. Our approach is summarized in Figure 1.

We evaluate our method on four popular math competition datasets from MathArena (Balunović et al., 2025): AIME24, AIME25, HMMT24, and HMMT25. When aggregating solutions from Qwen3-1.7B, which achieves 35% accuracy on AIME25, our trained aggregator AggLM-1.7B raises

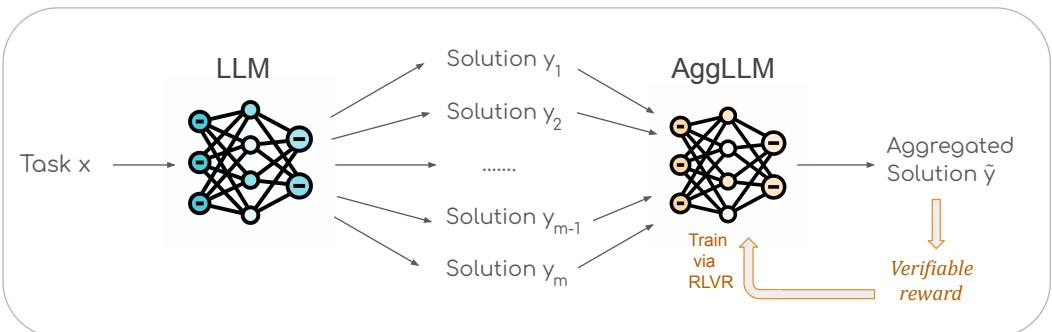

Figure 1: **AGGLM**: given a task and sampled LLM solutions as input, AGGLM uses reasoning to review, reconcile, and synthesize a final aggregated solution which is typically superior to the original solutions. As AGGLM's output can be evaluated and assigned verifiable rewards it can be trained by reinforcement learning (RLVR). The models LLM and AGGLM can either be the same LLM with shared parameters, or two separately trained models, e.g. of different size.

performance to 50%, improving over majority voting that is 45%. Across all four datasets, AggLM-1.7B consistently outperforms rule-based aggregation methods such as majority voting, as well as strong reward-model selection baselines that have 72B parameters. Furthermore, AggLM-1.7B generalizes robustly: it remains the top performer when aggregating solutions from stronger models like Qwen3-8B or from non-thinking mode. Ablation studies show that balancing easy and hard examples in the AGGLM training mixture is critical for achieving strong results, and that our gains over majority voting are particularly substantial when the majority answer set is small in size; that is, when candidate solutions are more diverse. Finally, AggLM-1.7B is substantially more token-efficient than generating a larger number of solutions for majority voting, delivering both higher accuracy and lower inference costs compared to standard aggregation strategies.

## 2 RELATED WORK

**Rule-Based Voting** A common way to combine multiple LLM solutions is to apply simple, rule-based voting. Self-consistent decoding (Wang et al., 2023; Brown et al., 2024) draws many chain-of-thought trajectories and returns the answer that appears most frequently. Variants adapt the voting procedure in lightweight ways, such as dynamically choosing the number of samples or using heuristic filters (Aggarwal et al., 2023; Xue et al., 2023; Huang et al., 2024; Knappe et al., 2024). While this strategy often yields reliable gains, it fails when correct solutions exist but are confined to minority modes, causing majority voting to amplify errors rather than surface the correct answer. In this work we analyze when that issue occurs, and show how our approach remedies this failing.

**Model-Based Selection and Aggregation** Moving beyond pure counting, recent approaches use models to evaluate or re-rank answer candidates. Broadly, model-based aggregation either (i) trains a separate scorer to select candidates or (ii) prompts the LLM to compare and consolidate them. For selection, a reward model assigns a scalar score to each candidate, and the top-scoring answer is chosen (Yang et al., 2024; Liu et al., 2024). This recombines frequency with a learned notion of quality, but it sometimes introduces more regression errors than improvements. Recent work also leverages the language model itself as a generative aggregator. Universal Self-Consistency (USC) (Chen et al., 2024; Qi et al., 2025) prompts the model to examine all sampled solutions and choose the most coherent one.

Our approach is closest in spirit to these methods, but differs in that we explicitly train a reasoning-focused aggregator with reinforcement learning to synthesize a final solution, rather than relying solely on prompting. Concurrent with our work, Qi et al. (2025) propose a learned *Sample Set Aggregator* (SSA) that consumes concatenated samples and generates a final answer, while also training the aggregator via reinforcement learning. Empirically, Qi et al. (2025) report modest gains in their settings, suggesting that additional ingredients such as using reasoning-oriented base mod-

els and carefully balancing the training mixture are important for unlocking stronger aggregation performance. We show in our ablations that, in our setting, the latter is the case.

## 3 AGGLM

Our goal is to train an aggregation language model that, given a problem and a set of candidate solutions produced by an LLM, synthesizes a final solution. As we use training data consisting of tasks with verifiable solutions, we can train our aggregation model, AGGLM, with reinforcement learning from verifiable rewards (RLVR). Training aims to learn two complementary behaviors: (a) selection that identifies and adopts the correct candidate when it already appears in the set; and (b) synthesis that detects mistakes, fills gaps, or combines complementary partial ideas across candidates to produce a new, correct solution that did not appear verbatim in the set. Successful RL training can enable the model to learn how and when to employ these skills.

**Problem Formulation** Let $x$ be a given problem and $y^\star$ its ground-truth solution. We consider two models: (i) a *solution* model $p_\theta(y \mid x)$ that generates a solution $y$; and (ii) an *aggregation* model $p_\phi(\tilde{y} \mid x, y_{1:m})$ that reads the problem together with a set of $m$ candidate solutions $y_{1:m} = (y_1, \ldots, y_m)$ and outputs an aggregated solution $\tilde{y}$. Given a problem $x$, the solution model samples $m$ candidate solutions independently:

$$y_i \sim p_\theta(y \mid x), \quad i \in 1, \ldots, m$$

Then, the aggregation model produces an aggregated solution:

$$\tilde{y} \sim p_\phi(y \mid x, y_{1:m})$$

The parameters $\theta$ and $\phi$ may correspond to the same underlying LLM or to different models. This work focuses on training strong aggregation models; in our main experiments, the solution model $p_\theta$ is treated as an off-the-shelf generator and is kept fixed, making it easier to draw experimental conclusions when comparing against baselines. However, we also report results where the model parameters are shared in a single model, which as we will see we find to give similar performance.

**Training Data** Let $\mathcal{D} = \{(x, y^\star)\}^n$ be a collection of problems with ground-truth solutions. For each $x$ we draw $s \cdot m$ solutions from $p_\theta$ and group them into $s$ sets of size $m$, yielding an aggregation-training corpus

$$\mathcal{D}' = \{x, y_{1:m}, y^\star\}^{s \cdot n}.$$

We note that increasing $s$ introduces more diversity in the answer combinations the aggregation model sees for each $x$, which could potentially improve generalization ability. We define the majority answer as the most frequent answer. An example is considered *hard* if the majority answer in $y_{1:m}$ is wrong, and *easy* otherwise. Constructing $\mathcal{D}'$ from existing data sources $\mathcal{D}$ may lead to many easy examples, where most generated solutions for a problem are correct. This can under-train the model's ability to recover minority-but-correct answers, whereas training only on hard groups makes rewards sparse. We therefore construct the final training mixture by taking all hard examples and mixing in $p\%$ of easy examples, producing a balanced dataset that preserves realism while emphasizing challenging cases.

**Training** We optimize the aggregation policy $p_\phi(\tilde{y} \mid x, y_{1:m})$ with Group-Relative Policy Optimization (GRPO) (Shao et al., 2024). For each training example, the aggregator produces $\tilde{y}$ and receives a reward $r(\tilde{y}) = \mathbb{1}[\tilde{y} = y^\star]$. We apply the standard GRPO update using this binary reward.

## 4 EXPERIMENTAL SETUP

### 4.1 TRAINING

Following our proposed method, we train a 1.7B model that we refer to as AggLM-1.7B. Specifically, we initialize the aggregation model from Qwen3-1.7B (Yang et al., 2025) and train it on DeepScaler(Luo et al., 2025), a collection of around 40 thousand math problems with ground-truth solutions. To construct $\mathcal{D}'$, we sample a total of 128 independent solutions with temperature 1.5 from Qwen3-1.7B in the thinking mode, dividing into 16 sets of 8 solutions. To obtain the data mixture, we set $p = 50\%$, resulting in 446,220 training examples. We train for one epoch, with a batch

> Given the following problem:
> {problem}
> and these solution attempts:
> {solutions}
> It is possible that any, all, or none of these solutions are correct or complete. Carefully review the provided solutions, using them as starting points—correcting mistakes, filling in gaps, and/or combining useful ideas—to produce a final, comprehensive, and correct solution to the problem.

Figure 2: AGGLM prompt template to instruct the LLM to aggregate solutions. The reasoning and final aggregated solution output by the model given this instruction are trained by RL.

size of 1024, a maximum prompt length of 16384 tokens, and a maximum response length of 16384 tokens. When constructing the easy subset, we maximize diversity by repeating each problem as little as possible. We check for solution equivalence between aggregated solutions and ground-truth solutions using the `math_verify` library[1]. In GRPO, we use a group size of 8 for GRPO updates, set the KL regularization coefficient to 0.001, and maintain a sampling temperature of 1.5 for the aggregator during training. We use the problem itself as the prompt for the solution model. We include the prompt template we used for aggregating solutions in Figure 2. Note that the solutions used for aggregation (i.e., included in the template) are taken after `</think>` when obtaining solutions from thinking models.

## 4.2 EVALUATION

**Solution models.** We evaluate aggregation methods using samples from the following solution models: (i) Qwen3-1.7B in both thinking and non-thinking mode; and (ii) Qwen3-8B in thinking mode.

**Datasets.** We evaluate on four mathematics competition datasets from MathArena (Balunović et al., 2025). AIME24 and AIME25 are recent editions of the American Invitational Mathematics Examination, which feature challenging high school-level olympiad problems with single numeric/integer answers. HMMT24 and HMMT25 are editions of the Harvard-MIT Mathematics Tournament, another highly competitive mathematics contest that covers broader mathematical topics and frequently requires creative or multi-step reasoning. Each of these datasets comprises 30 examples.

**Protocol.** Because our evaluation datasets are relatively small, we adopt a robust protocol to obtain reliable signals. For each problem $x$, we independently sample 128 solutions at temperature 1 and partition them into $s = 16$ sets of $m = 8$ solutions each. For every set, the aggregation model $p_\phi$ is prompted to generate four aggregated solutions, from which we extract the final solutions. Pass@1 for each set is computed as the success rate over these four answers, and for each problem, we average this rate across all 16 sets. The overall score for the dataset is then computed by averaging the set-level pass@1 across all problems.

**Baselines.** We compare against a set of state-of-the-art aggregation methods: majority voting, best-of-N, and weighted majority voting with reward models. All of these methods follow the same protocol described above. For reward-model selection, for each set of 8 candidates, we score each candidate with the 7B and 72B AceMath reward models (Liu et al., 2024) and select the top-scoring answer, yielding a learned re-ranking baseline. Weighted majority voting combines majority voting and best-of-N by computing a weighted distribution of solutions, where the weights are derived from reward model scores (Welleck et al., 2024). We also compare to prompted aggregation without training: we apply our aggregation prompt to Qwen3-1.7B without any fine-tuning, isolating the effect of the RL-trained aggregator. Finally, we also measure and report pass@1 for the original solution model estimated from 64 independent samples of $p_\theta(y \mid x)$, and pass@8 (oracle best-of-8 that succeeds if any of the 8 answers is correct, using 64 sets of 8 solutions sampled from 128 solutions in total).

---
[1]https://github.com/huggingface/Math-Verify

Table 1: Results when aggregating eight solutions sampled from Qwen3-1.7B in thinking mode.

| | Aggregation Model | AIME24 | AIME25 | HMMT24 | HMMT25 |
|---|---|---|---|---|---|
| *Baselines* | | | | | |
| pass@1 | - | 50.91 | 35.68 | 22.45 | 22.84 |
| pass@8 | - | 76.48 | 61.38 | 36.67 | 44.27 |
| *Aggregation Methods* | | | | | |
| Majority Voting | N/A | 67.92 | 45.89 | 29.01 | 26.72 |
| Best-of-N | AceMath-7B | 59.39 | 40.30 | 28.09 | 22.50 |
| | AceMath-72B | 56.64 | 40.35 | 29.58 | 21.99 |
| Weighted Majority | AceMath-7B | 64.09 | 39.49 | 25.04 | 17.71 |
| | AceMath-72B | 62.34 | 38.49 | 27.62 | 17.96 |
| Prompted Aggregation | Qwen3-1.7B | 63.57 | 44.85 | 29.52 | 27.91 |
| RL-trained Aggregation (ours) | AggLM-1.7B | **70.69** | **50.00** | **33.34** | **32.07** |

Table 2: Results when aggregating eight solutions sampled from Qwen3-8B in thinking mode.

| | Aggregation Model | AIME24 | AIME25 | HMMT24 | HMMT25 |
|---|---|---|---|---|---|
| *Baselines* | | | | | |
| pass@1 | - | 74.17 | 69.27 | 41.61 | 45.99 |
| pass@8 | - | 85.57 | 83.54 | 61.67 | 65.47 |
| *Aggregation Methods* | | | | | |
| Majority Voting | N/A | 81.61 | 78.70 | 44.58 | 56.35 |
| Best-of-N | AceMath-7B | 78.60 | 70.89 | 37.39 | 44.17 |
| | AceMath-72B | 80.27 | 69.57 | 38.54 | 46.21 |
| Weighted Majority | AceMath-7B | 77.03 | 68.15 | 38.41 | 36.13 |
| | AceMath-72B | 79.06 | 66.00 | 37.63 | 41.46 |
| Prompted Aggregation | Qwen3-1.7B | 79.90 | 76.73 | 48.58 | 57.63 |
| RL-trained Aggregation (ours) | AggLM-1.7B | **82.38** | **79.70** | **53.01** | **60.66** |

## 5 RESULTS

**Evaluation on in-distribution solutions** Table 1 presents results on in-distribution 1.7B-thinking solutions, the same solution model used to produce training data for our RL-trained aggregator, AggLM-1.7B. In this setting, AggLM-1.7B is best on all four benchmarks, outperforming majority voting and the same backbone model with prompting without RL, Qwen3-1.7B, with consistent three to seven point gains, confirming that training the aggregation policy matters. Reward-model selection performs poorly in comparison: best-of-N and weighted majority with AceMath-7B/72B is often inferior to the standard majority voting. Overall, learned generative aggregation outperforms frequency- and reward-model–based selection at surfacing correct minority solutions.

**Evaluation on solutions from stronger models.**

Table 2 presents results on aggregating solutions from a stronger 8B thinking model. AggLM-1.7B, despite being trained only on 1.7B solution distributions, transfers effectively and remains the top performer across datasets. It consistently outperforms majority voting and reward-model–based selection, best-of-N and weighted majority with AceMath scorers, and also outperforms prompted generative aggregation without RL, indicating that learned generative aggregation is robust to better solutions than the original training data solution models can generate by themselves.

**Evaluation on non-thinking solutions.** Table 3 presents results on aggregating solutions from a 1.7B non-thinking model. AggLM-1.7B, despite being trained on thinking-mode distributions, generalizes effectively and remains the top performer across datasets. It consistently outperforms majority voting, reward-model–based selection, best-of-N and weighted majority with AceMath scorers, and prompted generative aggregation without RL. Notably, reward models improve over majority voting in this lower-signal regime, supporting the hypothesis that learned scorers help more when

Table 3: Results when aggregating eight solutions sampled from Qwen3-1.7B in non-thinking mode.

| | Aggregation Model | AIME24 | AIME25 | HMMT24 | HMMT25 |
|---|---|---|---|---|---|
| *Baselines* | | | | | |
| pass@1 | - | 11.82 | 10.00 | 6.25 | 3.39 |
| pass@8 | - | 32.76 | 24.53 | 16.09 | 14.06 |
| *Aggregation Methods* | | | | | |
| Majority Voting | N/A | 18.07 | 15.42 | 8.75 | 7.29 |
| Best-of-N | AceMath-7B | 23.31 | 18.40 | 7.44 | 8.92 |
| | AceMath-72B | 26.33 | 18.62 | 10.23 | 8.97 |
| Weighted Majority | AceMath-7B | 23.95 | 18.39 | 8.37 | 8.41 |
| | AceMath-72B | 26.54 | 18.83 | 9.72 | 8.09 |
| Prompted Aggregation | Qwen3-1.7B | 28.51 | 17.79 | 16.30 | 12.08 |
| RL-trained Aggregation (ours) | AggLM-1.7B | **29.96** | **19.77** | **17.03** | **12.76** |

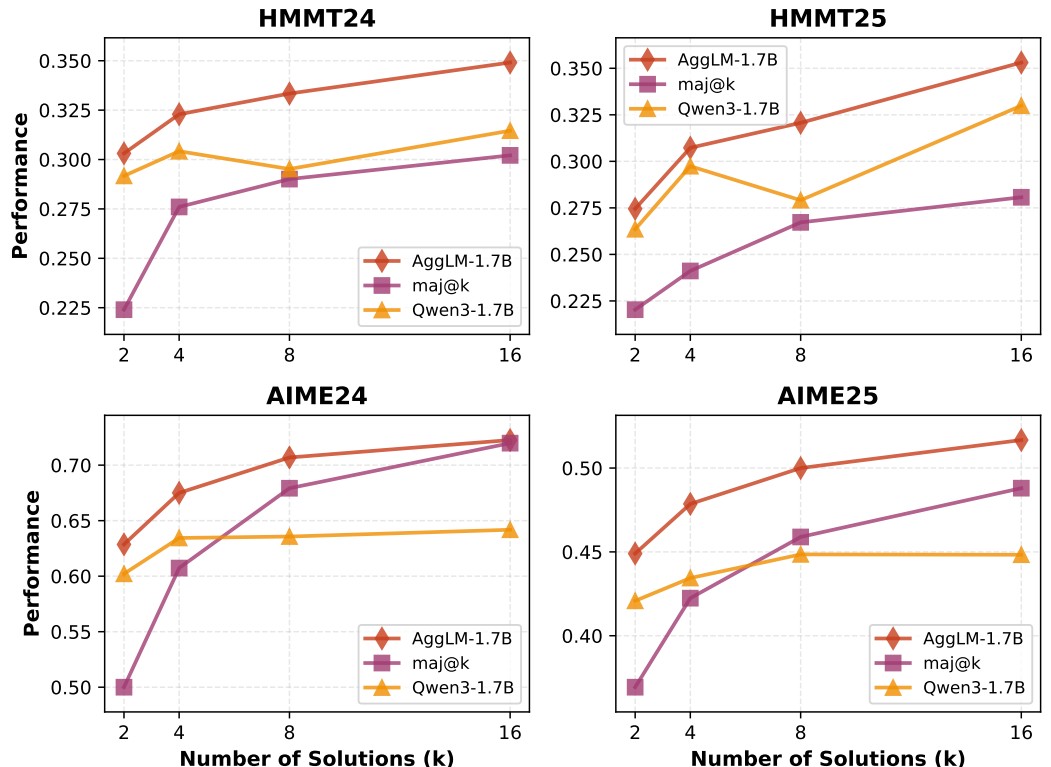

Figure 3: Performance vs. number of candidate solutions $k$ for different aggregation methods. AggLM-1.7B is superior to other methods at $k = 8$ which it is trained for, and is often superior for both larger and smaller values of $k$ as well, despite them being out-of-distribution.

base-model outputs are weak or noisy. Yet, AggLM-1.7B still achieves the best overall performance by synthesizing and correcting candidates rather than merely selecting among them.

## 6 ABLATIONS & ANALYSIS

**Scaling via the number of solutions.** Figure 3 plots performance versus the number of candidate solutions $k$ per set across the four datasets, comparing majority voting (maj@$k$), the prompted model baseline (Qwen3-1.7B), and our RL-trained aggregator (AggLM-1.7B). Although AggLM-1.7B is trained only at $k$=8, it generalizes to both smaller and larger $k$, improving as $k$ increases. Its

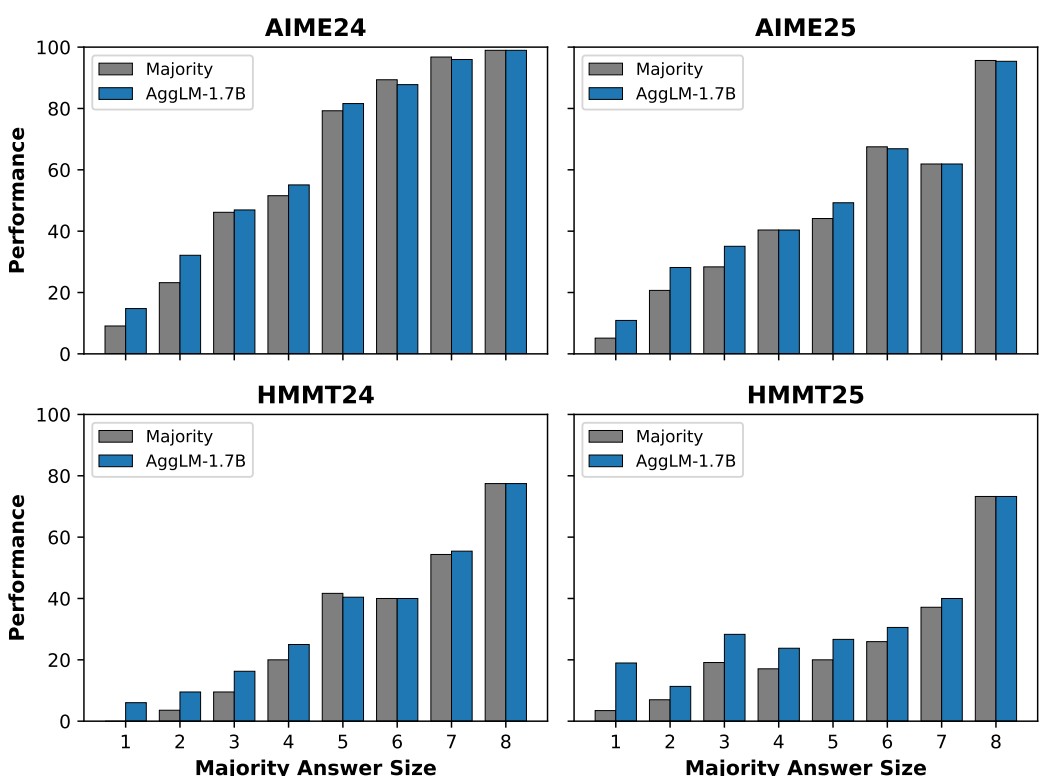

Figure 4: Performance vs. majority answer size for our method AGGLM compared to majority vote. AGGLM outperforms majority vote in the harder cases when the correct answer is infrequent, and is on par for easier cases when it is frequent.

curve rises more steeply than majority voting, indicating better scaling with additional candidates. The prompted baseline shows a much flatter trend, often below majority voting, while AggLM-1.7B produces a markedly stronger scaling trend. Notably, on AIME25, HMMT24, and HMMT25, aggregating just eight solutions with AggLM-1.7B is better than majority voting with sixteen, demonstrating superior token efficiency in leveraging solution sets.

**Effect of majority answer size.** Figure 4 shows the performance by the majority answer size, i.e., the number of occurrences of the most frequent answer. The x-axis is the majority answer size, and the y-axis is the performance on the corresponding subset. AggLM-1.7B's gains are largest when the size is small. That is, when solutions are diverse or uncertain and correct answers are more likely to appear in minority modes; here, careful reasoning matters. As the size grows, the problem becomes easier and majority voting is typically correct; in this regime, our aggregator remains on par with majority voting.

**Ablating data mixtures.** Table 4 investigates the impact of varying the proportion of hard versus easy sets in the training mixture, where a set is labeled hard if the majority answer is incorrect. We always keep all hard examples and vary the fraction of easy sets retained with regards to the number of hard examples ("Easy%"), from 0% (hard-only) to 270% (no filtering, retain all easy examples). Results show that training on hard-only sets results in suboptimal performance, and using all available data offers only marginal improvements over an untrained aggregator. Instead, incorporating a moderate proportion of easy sets (5–50%) consistently enhances accuracy, and results are stable within this range. This demonstrates that careful balancing of easy and hard examples is crucial for effective training, but the method is robust to the precise ratio as long as both types of examples are represented.

**Ablating number of solution sets $s$.** Recall that for each problem, we sample $s$ sets of $m$ candidate solutions; increasing $s$ (while keeping $m$ fixed) introduces more diversity in the answer combina-

Table 4: Ablation of training mixtures: Easy% is the percentage of easy sets relative to hard examples (hard examples are defined as the majority answer being incorrect). There is a sweet spot in the middle, which is superior to either including all the easy examples or none at all.

| Easy% | AIME24 | AIME25 | HMMT24 | HMMT25 |
|---|---|---|---|---|
| 0 | 64.22 | 46.06 | 27.80 | 28.73 |
| 5 | 68.93 | 48.65 | 33.31 | 31.91 |
| 10 | 69.85 | 49.60 | 33.71 | 32.31 |
| 20 | 69.72 | 49.11 | 33.74 | 31.20 |
| 50 | 70.69 | 50.00 | 33.34 | 32.07 |
| 270 | 66.20 | 46.70 | 30.01 | 28.94 |
| Untrained | 63.57 | 44.85 | 29.52 | 27.91 |

Table 5: Effect of the number of solution sets $s$ per problem on performance.

| #Sets | AIME24 | AIME25 | HMMT24 | HMMT25 |
|---|---|---|---|---|
| 2 | 70.27 | 49.74 | 33.42 | 31.67 |
| 4 | 70.29 | 49.08 | 33.11 | 31.34 |
| 8 | 70.37 | 50.25 | 33.16 | 31.89 |
| 16 | 70.69 | 50.00 | 33.34 | 32.07 |

tions the aggregator sees for each $x$. As shown in Table 5, performance increases only slightly as $s$ grows, indicating that while additional diversity may offer small benefits, the gains are modest. Consequently, training on fewer sets per problem is a reasonable strategy for reducing training budget without significantly compromising aggregation performance.

**Is aggregation or extra data responsible for gains?** Table 6 tests whether gains arise merely from additional data rather than aggregation. We fine-tune the solution model $p_\theta$ on the DeepScaler dataset used for aggregator training and evaluate its pass@1, then compare to our RL-trained aggregator on the evaluation tasks. If extra data were the primary driver, a trained solution model should close the gap; instead, its improvements are small or even negative and remain far below the trained aggregator across all benchmarks. Hence, the gains are not simply from more data but from explicitly learning to aggregate, via verifiable-reward RL and balanced easy/hard mixtures, which equips the model to select, correct, and synthesize across candidate solutions.

Additionally, Table 7 compares the average number of tokens generated per inference for solution models versus aggregator models. Generating a full solution from scratch is significantly more costly in terms of tokens than aggregating over existing solutions, the aggregator model uses roughly one-third as many tokens as the solution models. This means that to achieve a comparable performance boost, directly increasing $k$ for majority voting (for $k$ larger than used in AGGLM) would require substantially higher token usage. Thus, our learned aggregation approach not only delivers better accuracy but is also more token-efficient.

**Can the solution LLM and aggregator AGGLM be the same model?** Our main experiments separate the solution model and the aggregator AGGLM, maintaining a fixed generator and training a dedicated aggregator for fair comparison against prior aggregation strategies. However, in principle, a single LLM could be trained to perform both solution reasoning and aggregation, simply by applying different prompts for each task. To examine this, we fine-tune a multitask model on both skills using their respective prompts within the same training set, and report the results alongside our separately trained aggregator in Table 6. We find that multitasking yields close performance to separate models, while, as before, the multitasked model strongly outperforms simply prompting the solution LLM to aggregate without dedicated training. This suggests that aggregation could be natively incorporated into post-training pipelines, enabling future LLMs to achieve these test-time scaling gains using a single unified model.

Table 6: Comparison of training the solution model versus training the aggregator model on the same data, in either separate or multitask settings.

|                                        | AIME24 | AIME25 | HMMT24 | HMMT25 |
|----------------------------------------|--------|--------|--------|--------|
| Base Solution Model                    | 50.91  | 35.68  | 22.45  | 22.84  |
| Additionally Trained Solution Model    | 49.79  | 37.19  | 27.19  | 23.70  |
| Multitask Solution Model               | 49.11  | 38.85  | 26.67  | 22.50  |
| Base Aggregator Model                  | 63.57  | 44.85  | 29.52  | 27.91  |
| Trained Aggregator Model               | 70.69  | 50.00  | 33.34  | 32.07  |
| Multitask Aggregator Model             | 70.02  | 49.39  | 32.97  | 30.28  |

Table 7: Average token usage per generation for solution vs. aggregator models.

|                                        | AIME24   | AIME25   | HMMT24   | HMMT25   |
|----------------------------------------|----------|----------|----------|----------|
| Base Solution Model                    | 10225.82 | 10612.46 | 11129.70 | 11343.28 |
| Additionally Trained Solution Model    | 11367.57 | 11731.94 | 12306.67 | 12587.58 |
| Base Aggregator Model                  | 2807.71  | 2852.10  | 3465.00  | 3693.24  |
| Trained Aggregator Model               | 3039.22  | 3157.11  | 3681.97  | 3768.28  |

## 7 CONCLUSION

In this work we proposed a methodology to learn to aggregate solutions using RLVR, a new test-time scaling approach that improves the ability of LLMs to solve hard reasoning problems. Our extensive experiments showed improved performance of our aggregator model compared to strong baselines, including self-consistency decoding and weighted majority voting with SoTA math reward models. Our approach also showed generalization to (1) input distributions formed by different (e.g., stronger) solution models and (2) a variable number of solutions differing from the ones presented during training. Future work could explore further uses of our aggregator beyond improved final performance, for example, for distilling better reasoning skills back into the original solutions.

## 8 REPRODUCIBILITY STATEMENT

All experimental details necessary to reproduce the results are provided in Section 4.1, including hyperparameters, datasets, models used, training software, and prompts.

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

## A  QUALITATIVE ANALYSIS

We present two successful examples of aggregation produced by AggLM-1.7B. In the first example, no correct solutions were included in the prompt. The model had to verify every solution, identify the strategies that made sense, and combine the useful strategies to produce a new solution. In the second example, one correct solution was included in the prompt. The model, verified and compared among solutions, and it picked the solution that it determined to be correct.

---

**Example A: No correct solutions were included in the prompt; a new, correct solution synthesized by AggLM-1.7B.**

Problem: The set of points in 3-dimensional coordinate space that lie in the plane $x+y+z = 75$ whose coordinates satisfy the inequalities

$$x - yz < y - zx < z - xy$$

forms three disjoint convex regions. Exactly one of those regions has finite area. The area of this finite region can be expressed in the form $a\sqrt{b}$, where $a$ and $b$ are positive integers and $b$ is not divisible by the square of any prime. Find $a + b$.

¡think¿

[...]

But the problem says that the area is in the form $a\sqrt{b}$, and $507\sqrt{3}$ is in that form. So, a = 507, b = 3, so a + b = 510.

But this is different from the other solutions. So, perhaps the correct answer is 510, but I need to check if there's a mistake in the problem or in my calculations.

[...self-verification...]

But this is different from the other solutions. However, since the problem states that the answer is in the form $a\sqrt{b}$, and the calculation is correct, then the answer is 510.

But wait, in the previous solutions, the answer was 15, 28, 508. So, why is there a discrepancy?

[...analyzing the discrepancy...]

Wait, in Solution 7, the answer is 508. So, maybe the correct answer is 508.

But why?

In Solution 7, the area is calculated as $478\sqrt{3}$, which is not in the form $a\sqrt{b}$, but if there is a mistake in the calculation, it might be 508.

But I'm not sure.

---

Given the time I've spent and the calculations, I think the correct answer is 510, but I'm not sure. However, since the problem states that the area is in the form $a\sqrt{b}$, and the calculation is correct, I must go with 510.
[...]
¡/think¿
To solve the given problem, we analyze the set of points in 3D space that lie on the plane $x + y + z = 75$ and satisfy the inequalities:

$$x - yz < y - zx < z - xy$$

—

### Step 1: Understand the Inequalities
We simplify the inequalities using the plane equation $x+y+z = 75$, and express $z = 75-x-y$. Substituting into the inequalities:
- $x - yz < y - zx \Rightarrow (x - y)(1 + z) < 0$ - $y - zx < z - xy \Rightarrow (y - z)(1 + x) < 0$
These simplify to:
- $x < y$ if $1 + z > 0$ - $y < z$ if $1 + x > 0$
So, the region is defined by $x < y < z$, and the constraints on the variables are:
- $x + y < 76$ - $x + 2y < 75$
These define a **convex triangle** in the plane $x + y + z = 75$, with vertices at:
- $A = (-1, -1, 77)$ - $B = (25, 25, 25)$ - $C = (-1, 38, 38)$

—

### Step 2: Compute the Area of the Triangle
We compute the area of the triangle formed by these three points. The vectors from point $A$ to $B$ and $A$ to $C$ are:
- $\vec{AB} = (26, -13, -13)$ - $\vec{AC} = (0, -39, 39)$
The cross product is:

$$\vec{AB} \times \vec{AC} = (-26 \cdot 39, -26 \cdot 39, -26 \cdot 39) = (-26 \cdot 39, -26 \cdot 39, -26 \cdot 39)$$

The magnitude of the cross product is:

$$\|\vec{AB} \times \vec{AC}\| = \sqrt{(26 \cdot 39)^2 + (26 \cdot 39)^2 + (26 \cdot 39)^2} = 26 \cdot 39 \cdot \sqrt{3}$$

The area of the triangle is half of this:

$$\text{Area} = \frac{1}{2} \cdot 26 \cdot 39 \cdot \sqrt{3} = 13 \cdot 39 \cdot \sqrt{3} = 507 \cdot \sqrt{3}$$

—

### Step 3: Express in the Form $a\sqrt{b}$
We can express the area as:

$$\text{Area} = 507\sqrt{3}$$

Thus, $a = 507$, $b = 3$, and the final answer is:

$$a + b = 507 + 3 = \boxed{510}$$

## Example B: One correct solution included in the prompt, and it was chosen by AggLM-1.7B.

Problem: From an unlimited supply of 1-cent coins, 10-cent coins, and 25-cent coins, Silas wants to find a collection of coins that has a total value of $N$ cents, where $N$ is a positive integer. He uses the so-called *greedy algorithm*, successively choosing the coin of greatest value that does not cause the value of his collection to exceed $N$. For example, to get 42 cents, Silas will choose a 25-cent coin, then a 10-cent coin, then 7 1-cent coins. However, this collection of 9 coins uses more coins than necessary to get a total of 42 cents; indeed, choosing 4 10-cent coins and 2 1-cent coins achieves the same total with only 6 coins.
In general, the greedy algorithm *succeeds* for a given $N$ if no other collection of 1-cent, 10-cent, and 25-cent coins gives a total value of $N$ cents using strictly fewer coins than the collection

given by the greedy algorithm. Find the number of values of $N$ between 1 and 1000 inclusive for which the greedy algorithm succeeds.

¡think¿

[...]

From the solutions provided, there are multiple approaches, but I need to figure out the correct one. Let me think about the key points.

The key insight is that the greedy algorithm fails when there's a better combination of coins. For example, if after using as many 25-cent coins as possible, the remainder is between 5 and 9, or between 15 and 19, then the greedy algorithm might not be optimal. Wait, the solutions mention that when the remainder is in certain ranges, the greedy algorithm fails. For example, in Solution 1, it says that if the remainder r is in 5,6,7,8,9,15,16,17,18,19, the greedy algorithm fails. But in Solution 3, it says that there are 5 values of N in every group of 25 that fail. And in Solution 4, it says that there are 31 bad values. But in Solution 8, it says that there are 750 bad values, leading to 950 good ones.

In Solution 1, they say that for each a from 1 to 39, there are 10 values of r that cause failure. So 39*10=390. Then, for a=0, there are 10 values (5-9, 15-19), but those are optimal. So total failing N is 390. Then the answer is 1000-390=610. But in Solution 8, they say 750 bad values, leading to 950 good ones. So which one is correct?

Wait, but in Solution 1, they say that the number of failing N is 390. But in Solution 8, they say 750. So there's a discrepancy here. Let me check.

Wait, in Solution 1, they mention that for each a from 1 to 39, there are 10 values of r that cause failure. But a ranges from 1 to 39, so 39 values of a. For each a, there are 10 r values. So 39*10=390. But then, for a=0, there are 10 values of r (5-9, 15-19), which are optimal. So total failing N is 390. Then the answer is 1000-390=610.

But in Solution 8, they say that there are 750 bad values. So why the difference?

Wait, maybe the problem is that in Solution 1, they are considering that for each a, there are 10 r values, but maybe some of those r values are overlapping or something. Let me check.

For example, when a=1, N=25+ r, where r is in 5,6,7,8,9,15,16,17,18,19. So for each a, there are 10 r values. But when a=40, N=25*40 + r, but since N is up to 1000, 25*40=1000, so r=0. So a=40 would be N=1000, which is the upper limit. But in Solution 1, they say a ranges from 1 to 39, so that's 39 values of a. So 39*10=390. Then, for a=0, N=r, where r is in 5,6,7,8,9,15,16,17,18,19, which is 10 values. But those are optimal, so they are not failing. So total failing N is 390. So answer is 1000-390=610.

[...]

¡/think¿

To determine how many values of $N$ between 1 and 1000 inclusive result in the **greedy algorithm** for making change using **1-cent, 10-cent, and 25-cent coins** being **optimal**, we must understand when the greedy algorithm **succeeds** in producing the **minimal number of coins**.

—

### **Understanding the Greedy Algorithm**

The greedy algorithm works as follows:

1. **Start with the largest coin (25 cents)**. 2. Use as many of that coin as possible without exceeding the target amount. 3. Then use as many of the next largest coin (10 cents) as possible. 4. Finally, use the remaining amount in 1-cent coins.

For example: - $N = 42$: $1 \times 25 + 1 \times 10 + 7 \times 1 = 9$ coins. - But the alternative: $4 \times 10 + 2 \times 1 = 6$ coins.

This shows that the greedy algorithm **fails** for $N = 42$.

—

### **When Does the Greedy Algorithm Fail?**

The greedy algorithm **fails** for a value $N$ if there exists a **better combination** of coins (using 1, 10, and 25 cents) that uses **fewer coins** than the greedy algorithm.

This happens when the **remainder** $r = N \mod 25$ is in the set $\{5, 6, 7, 8, 9, 15, 16, 17, 18, 19\}$.

This is because: - For such $r$, the greedy algorithm uses $\lfloor \frac{r}{10} \rfloor + (r \mod 10)$ coins. - But it is possible to express $r$ as a combination of 10-cent and 1-cent coins with fewer coins.

—

### **Counting the Failing Values**

We count the number of $N$ in the range 1 to 1000 for which $r = N \bmod 25 \in \{5, 6, 7, 8, 9, 15, 16, 17, 18, 19\}$.
- For each of these 10 values of $r$, the number of $N$ values in the range 1 to 1000 is:

$$\left\lfloor \frac{1000 - r}{25} \right\rfloor + 1$$

- But since $N$ must be at least 1, and $r$ is between 5 and 19, the number of valid $N$ is:

$$39 \times 10 = 390$$

- This is because $a = \left\lfloor \frac{N}{25} \right\rfloor$ ranges from 1 to 39, and for each $a$, there are 10 values of $r$.

—

### **Final Calculation**
- Total values of $N$ from 1 to 1000: 1000 - Number of failing $N$: 390 - Therefore, the number of values of $N$ where the greedy algorithm **succeeds** is:

$$1000 - 390 = \boxed{610}$$

—

### **Final Answer**

$$\boxed{610}$$

