# OpenReview forum: "The Majority is not always right: RL training for solution aggregation"
_ICLR.cc/2026/Conference — Submitted to ICLR 2026_

### Official Review · Reviewer_qX3S · 2025-10-30

**Soundness:** 3
**Presentation:** 4
**Contribution:** 3
**Rating:** 4
**Confidence:** 2

**Summary:**

This paper proposes a reinforcement learning (RL)-based method for large language models (LLMs) that performs aggregation of multiple sampled responses as part of test-time scaling. Unlike previous approaches that rely on naive frequency-based voting or reward-model ranking, the proposed method trains an aggregator model that generates a final answer by reasoning over all sampled responses.
This allows the model to handle diverse situations, such as when the correct answer is in the minority or when partially valid reasoning exists in incorrect responses. Through extensive experiments, the authors demonstrate the effectiveness of their approach and show that it remains robust across various settings, including in-/out-of-domain generalization, different numbers of candidate responses, and varying mixtures of training data.

**Strengths:**

1. The proposed method is **conceptually simple yet delivers clear performance improvements.** Notably, it can be trained using existing and publicly available response data through standard RL techniques, which suggests that it can serve as a practical foundation for future research in this direction.
2. The authors present **comprehensive robustness analyses.** They evaluate the method across diverse datasets, model configurations, answer sizes, and training data mixtures, demonstrating that the approach is consistently robust to these factors. As discussed in the conclusion, this robustness implies that the method could be integrated into post-training pipelines as a promising component for future test-time scaling strategies.
3. The paper employs **strong and fair baselines.** The baselines include large reward models specialized for mathematical reasoning (7B and 72B), which are far more resource-intensive than the 1.7B aggregator model used in this work. Despite the smaller size, the proposed method achieves superior performance and remarkable token efficiency, showing it is both effective and efficient.

**Weaknesses:**

1. The experiments are limited to the mathematics domain (e.g., AIME datasets). It remains unclear whether the proposed RLVR framework would **generalize to other reasoning-intensive domains** such as coding, where verifiable signals are also available.
2. The training data, DeepScaler, is designed in an AIME-like style, which raises the possibility of **data leakage or overfitting to similar problem types.** Further validation or control experiments would strengthen the claim of generalization. Clarifying and addressing this potential data overlap is essential to ensure the validity of the reported improvements.

**Questions:**

1. How does the **aggregator model scale with size**? Since generative aggregation requires reasoning over multiple candidate solutions (and even synthesizing correct reasoning from entirely incorrect ones), it is reasonable to expect that larger models might exhibit stronger aggregation ability. Have the authors examined such scaling trends?
2. When scaling up the aggregator, **does the performance gap between prompted aggregation and RL-trained aggregation remain?** Table 1 and Table 2 suggest that the difference narrows as the solution model becomes stronger, implying that RL-based aggregation might be less beneficial when the base solutions are already of high quality.
3. Would similar performance be observed if the model were trained on **datasets unrelated to AIME** or the math domain?
4. Could the proposed method be extended robustly to **other domains,** such as coding or scientific reasoning tasks?

---

### Official Review · Reviewer_3Zxb · 2025-10-31

**Soundness:** 3
**Presentation:** 3
**Contribution:** 3
**Rating:** 4
**Confidence:** 3

**Summary:**

This paper proposes a reinforcement learning-based approach (RLVR) for aggregating multiple candidate solutions generated by large language models (LLMs) in mathematical reasoning tasks. The method aims to overcome the limitations of traditional majority voting strategies, such as ignoring correct minority solutions or failing to synthesize partially correct reasoning distributed across different candidates. Specifically, the paper trains a dedicated aggregator model that learns from verifiable rewards to synthesize, correct, or select among multiple candidate solutions. Experiments conducted on four challenging math competition datasets demonstrate that this approach consistently outperforms majority voting, reward model-based baseline methods such as AceMath, and naive generative aggregation methods across various aggregation scenarios.

**Strengths:**

1. The paper accurately identifies the shortcomings of existing majority voting aggregation strategies, such as potentially overlooking correct minority solutions and failing to integrate partial reasoning distributed across different candidate answers. It compellingly argues for the necessity of learning an aggregation method.

2. The method was evaluated on four challenging math competition datasets, and the effectiveness of AggLM was analyzed from multiple perspectives, supporting the paper's claims

**Weaknesses:**

1. All experiments are limited to mathematical tasks that have verifiable rewards. For non-RLVR tasks, the training approach of AggLM seems no longer applicable.

2. The RLVR experiments are restricted to the Qwen series. Demonstrating cross-series generalization capabilities beyond Qwen and comparisons with closed-source models like GPT would be beneficial.

**Questions:**

1. Could the authors elaborate on whether AggLM provides advantages over directly fine-tuning LLMs with RLVR in tasks such as mathematics, where rewards can be readily verified?

2. Figure 4 suggests that when the number of candidate solutions is large, the performance gap between AggLM and majority voting tends to decrease, while greater benefits are observed with fewer candidates. Under comparable resource consumption (or inference latency), might the strategy of “fewer candidate solutions + AggLM” be more beneficial than “more candidate solutions + majority voting”?

3. The current experimental comparison is primarily with model-based methods. Have the authors considered including self-certainty [1] and PiCSAR [2] in the evaluation? These additional baselines might help highlight the strengths of AggLM more clearly.

>[1]Zhewei Kang, Xuandong Zhao, and Dawn Song. Scalable best-of-n selection for large language models via self-certainty. ArXiv preprint, abs/2502.18581, 2025. URL https://arxiv.org/abs/2502.18581.

>[2]Leang, J. O. J., Zhao, Z., Gema, A. P., Yang, S., Kwan, W.-C., He, X., Li, W., Minervini, P., Giunchiglia, E., & Cohen, S. B. (2025). PiCSAR: Probabilistic Confidence Selection And Ranking for Reasoning Chains. arXiv. https://arxiv.org/abs/2508.2178

---

### Official Review · Reviewer_ouoR · 2025-10-31

**Soundness:** 2
**Presentation:** 2
**Contribution:** 2
**Rating:** 4
**Confidence:** 4

**Summary:**

This paper proposes AGGLM, an approach for aggregating multiple candidate solutions generated by large language models (LLMs), with a focus on challenging reasoning tasks, such as those in mathematics. The central idea is to explicitly train an aggregator model to review, reconcile, and synthesize a correct final answer from a set of generated solutions, framing aggregation itself as a learned skill via reinforcement learning from verifiable rewards.

Experiments are conducted across four benchmarks, with results showing that the learned aggregator consistently outperforms majority voting, reward model selection, and naive aggregation baselines. The method is further shown to generalize across solution sources and is more token-efficient than existing approaches.

**Strengths:**

1. The paper addresses a well-motivated gap in solution aggregation for LLM reasoning, where majority-based or naive aggregation often fails, especially when correct solutions are minority or require synthesis across candidates.

2. Results span four math competition datasets, with in-depth benchmarking against majority voting, best-of-N, weighted majority via reward models, and prompted LLM aggregation, as shown in Tables 1, 2, and 3.

3. The evaluation methodology is spelled out clearly (Section 4.2) so experiments can be reproduced and compared fairly.

**Weaknesses:**

1. Limited model diversity. The study only employs Qwen3 as the solution model, without exploring other representative LLMs such as Llama. Moreover, the experiments are restricted to Qwen3-1.7B, which is insufficient to demonstrate the generalizability of the proposed RL training data and methodology across different model architectures or scales.

2. Lack of ablation on aggregation models and training datasets. The work relies solely on the Qwen3-1.7B (Thinking mode) model to construct training data from the DeepScaleR dataset. However, the paper does not clearly justify the choice of this specific model configuration or dataset, nor analyze their impact on aggregation performance. Ablation studies on the aggregation model and data composition are missing.

3. Limited evaluation scope. The evaluation benchmarks focus exclusively on mathematical reasoning, overlooking other critical domains such as coding, instruction following, and general-purpose reasoning. This limits the conclusions regarding the method’s overall reasoning capabilities.

4. Experimental setup concerns. The number of solution samples used for aggregation is not clearly justified. It remains unclear whether the chosen number is sufficient or representative to yield convincing conclusions.

5. Limitations of using AceMath as the reward model. The paper lacks a detailed description of AceMath. Furthermore, in Tables 1 and 2, the performance of the Weighted Majority approach based on AceMath consistently underperforms the simple Majority Voting baseline, suggesting potential issues with the chosen reward model.

6. Robustness issues in AggLM-1.7B. As shown in Figure 4 (AIME24 and AIME25), AggLM-1.7B exhibits decreased aggregation accuracy when the majority answer size is large. This indicates that the model may not be robust even within groups where the majority solution is correct.

7. Unclear setup in Table 6. The experimental configuration in Table 6 lacks clarity—specifically, whether the fine-tuning method is SFT or RL. Additionally, the observed performance degradation of the “Additional Trained Solution Model” on the math dataset is not well explained.

**Questions:**

1. Failure Analysis: Are there specific case studies illustrating the typical failure patterns of AggLM-1.7B during aggregation?

2. Sensitivity to Data Mixing: How does the performance of AggLM vary when different levels of noise or mixing ratios are introduced into the training data?

3. Comparison with Prompted Aggregation: Compared to Prompted Aggregation, what are the concrete advantages of AggLM-1.7B? Could the authors provide illustrative examples to clarify their strengths and the necessity of model-based training?

4. Statistical Significance: Given the relatively small dataset size (30 examples per benchmark), do the authors report confidence intervals or statistical tests to support the claim of consistent and reliable improvements?

---

### Official Review · Reviewer_Rp8o · 2025-11-03

**Soundness:** 3
**Presentation:** 3
**Contribution:** 3
**Rating:** 6
**Confidence:** 4

**Summary:**

This paper reframes test-time aggregation as a trainable reasoning skill: an aggregator LLM is fine-tuned with verifiable rewards to read a small set of candidate solutions, reconcile errors, and produce a corrected final answer. Beyond simple majority vote or reward-model selection, the approach focuses on two behaviors—selecting a correct minority candidate when it exists and synthesizing a new solution from partially correct traces—trained with a carefully balanced mixture of “hard” (majority wrong) and “easy” (majority right) sets. Evaluations on MathArena’s AIME24/25 and HMMT24/25 show consistent gains over majority voting and reward-model re‑ranking, strongest when the majority is small; the method generalizes across solution models (1.7B → 8B, thinking/non‑thinking) and exhibits favorable token efficiency at typical k.

**Strengths:**

1. This proposed method has a clear and timely reframing of aggregation as a learned reasoning skill using verifiable rewards and a lightweight RL procedure. It has consistent improvements over majority voting and reward‑model selection across multiple math benchmarks, with the largest gains when the correct answer appears in minority modes.

2. This proposed method has a practical and well‑motivated training mixture that balances hard and easy sets, and ablations indicate robustness within a useful range of ratios. The experiment has demonstrated generalization across solution model strengths and modes, improving results even when the aggregator is trained on distributions from a smaller model.

3. This method has favorable token efficiency at common settings, as scaling curves suggest that aggregating a modest number of candidates can outperform majority voting with larger k.

**Weaknesses:**

1. Order/duplication sensitivity: The aggregator sees a sequence of candidates; report whether permutation of input order or deduplication of near‑identical solutions changes performance.

2. Metric clarity: The evaluation has a nonstandard “pass@1” definition that averages over four aggregated samples per set, which may hinder comparison to prior work using strict single‑sample pass@1.

3. The experiment has no confidence intervals or variance estimates on 30‑item datasets, so several 2–4 point gaps are difficult to assess statistically. Considering that this method could be sensitive to candidate order, one could randomize the order and re-evaluate for multiple times to get the statistics (e.g., confidence intervals or standard errors).

**Questions:**

1. The compute analysis has incomplete accounting because input token costs (prefilling) for concatenated candidates and the cost of generating those candidates (decoding) are not included in efficiency comparisons. Can authors provide some numbers?

2. Verifier sensitivity: How sensitive are results to Math‑Verify configuration? Any cases where it mis‑evaluated numerically equivalent forms? A small audited subset (human‑verified) would be reassuring.

---

### Official Review · Reviewer_BrS5 · 2025-11-05

**Soundness:** 3
**Presentation:** 3
**Contribution:** 3
**Rating:** 6
**Confidence:** 3

**Summary:**

The paper proposes a new approach to aggregate multiple outputs from an LLM to a final answer. Instead of using simple rules as in best-of-N or Self-consistency, an LLM is fine tuned by RLVR to read all generated outputs and write a new solution. With careful balancing of training data, the new approach outperforms the simple rule-based ones.

**Strengths:**

I like the idea of the paper to learn the aggregation step. With more and more aggregation algorithms being introduced recently, this might be a path to a more optimal solution.

The paper is written well and describes the details of the approach very well.

Reasonable ablations is done and nice insights are provided such as the comparison for each difficulty class.

Showing a small aggregation model can aggregate the outputs of a larger model was nice.

**Weaknesses:**

My main concern about the paper is that it has really limited (or perhaps no) discussions on the limitations of the approach. With such a drastic difference to traditional approaches, it is very useful to study the differences and limitations.

My suggestion is to challenge the method more. The most pressing question is how far can the number of aggregated outputs be increased. The paper provides results for up to 16 but I suspect there is ceiling. Presenting this ceiling is very valuable.

Also, I am curious to see how much the model generalizes to other topics. For example Chemistry questions might still work as they benefit from many steps and reasoning, but common sense may not.

It would also be nice to try this approach in open ended problems, for example code and writing. I wonder if composing different solutions into a coherent one is more difficult in such domains.

**Questions:**

See weaknesses

---

### Meta-Review · Area_Chair_PZqK · 2025-12-29

**Summary:**

The decision is informed by significant concerns regarding the method's limited generalization, insufficient experimental rigor, and unproven robustness. The system exhibits critical vulnerabilities regarding potential overfitting, sensitivity to input perturbations, and unverified scalability in expanded search spaces. Furthermore, the approach lacks empirical validation across diverse domains and architectures beyond specific mathematical tasks and a single model family.

**Reviewer Concerns:**

The authors did not provide any substantive rebuttal to the reviews. All identified concerns remain.

**Reviewer Scores:**

Given the lack of an effective and positive rebuttal, it is probable that the reviewers would lower their scores.

---

### Decision · Program_Chairs · 2026-01-26

Reject